# The Effectiveness of Paracetamol to Reduce the Post-Vaccination SARS-CoV-2 Adverse Effects in an Italian Vaccination Center

**DOI:** 10.3390/vaccines11091493

**Published:** 2023-09-15

**Authors:** Eleonora Ricci, Anamaria Glavasc, Barbara Morandini, Maria Caterina Grassi, Giuseppe La Torre

**Affiliations:** 1Department of Public Health and Infectious Diseases, Sapienza University of Rome, 00185 Rome, Italy; eleonora.ricci@uniroma1.it; 2San Giovanni Battista Hospital, 00148 Roma, Italy; glavasc.1532686@studenti.uniroma1.it (A.G.); morandini.684782@studenti.uniroma1.it (B.M.); 3Department of Physiology and Pharmacology “V. Erspamer”, Sapienza University of Rome, 00185 Rome, Italy; caterina.grassi@fondazione.uniroma1.it

**Keywords:** effectiveness, paracetamol, vaccination SARS-CoV-2, adverse effects, Italy

## Abstract

The arrival of specific vaccines was crucial for the eradication of COVID-19. Despite the security of the vaccination, the administration of COVID-19 mRNA vaccines often causes systemic side effects for a short time after the injection, such as headache, fatigue, fever, muscle pain and nausea. These side effects can limit the adherence to COVID-19 vaccines administration, especially in healthcare workers. This study aims to analyze the impact of the prophylactic use of paracetamol to reduce the post-vaccination Comirnaty/Pfizer adverse effects. The study took place at the San Giovanni Battista Hospital in Rome and included all hospital employees who received two doses of Pfizer-BioNTech. The vaccination health personnel recommended the preventive intake of 1 g of paracetamol before the inoculation of the vaccine and then every 6 h in the first 24 h. Information was collected through forms and subsequent telephone recall. A total of 403 volunteers were involved in the study, with 391 of them receiving two doses and twelve individuals only one dose of the vaccine. The main results demonstrated that the prophylactic therapy influenced the lower onset of asthenia in the first and second doses. However, there were no significant differences between the two groups in terms of fever, headache and localized pain. Paracetamol had a good impact on the side effect of COVID-19 vaccination, reducing asthenia in both doses and mitigating the total of symptoms during the second vaccination.

## 1. Introduction

COVID-19 has changed our lives since late December 2019 when, in China, a new betacoronavirus caused numerous pneumonia cases with no etiology [1]. Sars-Cov2 was identified as the agent responsible for coronavirus disease 2019 (COVID-19), an ongoing global pandemic declared a health emergency by the World Health Organization (WHO) in March 2020. COVID-19 is still raging worldwide, with 6.2 million deaths reported [2], including 160,000 in Italy alone [3].

Social distancing, the use of personal protection equipment (PPE) such as facemasks, and personal hygiene measures seemed to prevent the spread of SARS-CoV-2 in the community, albeit with some limitations. However, the arrival of specific vaccines was crucial for the eradication of COVID-19. In these two years, highly contagious variants have spread, but the two doses of vaccine are still considered effective in reducing the severity of the disease and mortality [4].

Despite the safety of vaccination, there is evidence that the administration of COVID-19 mRNA vaccines often causes systemic side effects for a short period after injection, including headache, fatigue, fever, muscle pain and nausea [5]. Occasionally, serious side effects have been reported, which can cause long-term health problems [6]. Nevertheless, when comparing the advantages and disadvantages of vaccination with mRNA vaccines, it has commonly been recommended. However, as the pandemic continues to evolve and be controlled, the sequelae of vaccination may become more evident. There are several hypotheses in this regard, such as an increase in cardiovascular diseases, specifically acute coronary syndromes linked to the spike proteins present in the vaccine, potential organ damage or an increased risk of infections due to reduced immune function [7].

Similar to the flu vaccination, these side effects can limit the adherence to COVID-19 vaccines administration, especially in healthcare workers [8]. In fact, as demonstrated by a cross-sectional survey of healthcare workers (HCWs) carried out in California, out the 2103 vaccinated, 34% experienced side effects during work, 28% had to interrupt their work, and 11% declared that the symptoms moderately interfered a strict requirement for the correct fulfillment of professional responsibilities. This can lead to absenteeism and impact future vaccine decision-making [9].

The use of some medicines, including paracetamol, can reduce or delay the side effects of vaccinations. In the USA, it is known as acetaminophen, and in Europe, it is called paracetamol, named after its chemical compound: N-acetyl-para-aminophenol. The WHO analgesics scale defines precisely the rules for the application of analgesic drugs and positions paracetamol in a favorable category. This drug is included in all three rungs of the pain scale, serving as a basic non-opioid analgesic. It is used either alone for mild pain or in addition to opioid medications for moderate to severe pain. Paracetamol is particularly recommended for long-term therapy, especially for conditions such as osteoarticular and muscular discomfort [9].

Although acetaminophen has broad clinical applications, it is not a drug without side effects. The main critical issue arising from the widespread use of paracetamol is its potential to cause hepatotoxicity in cases of overdose, as it undergoes metabolism into reactive compounds. Some studies claim that many cases of hepatotoxicity are associated with therapeutic doses of it, but a critical analysis indicates that many of these cases are the result of almost exclusively overdoses. The dosage of paracetamol, particularly in children and alcoholics, must be carefully controlled in order to prevent liver damage. Physicians should emphasize to patients the need to take the correct dosage of paracetamol [10].

Overall, paracetamol monotherapy is effective, well-tolerated by most patients, and safe, provided that the drug is administered in therapeutic doses.

In light of these considerations, the aim of this study was to analyze the impact of the prophylactic use of paracetamol to reduce the post-vaccination Comirnaty/Pfizer adverse effects in an Italian vaccination center.

## 2. Materials and Methods

### 2.1. Study Design and Setting

A non-randomized clinical trial study was carried out at the San Giovanni Battista Hospital in Rome. It included all hospital employees who received two doses of Pfizer-BioNTech; those who received a different vaccination were excluded. The healthcare personnel at the vaccination center recommended the preventive intake of 1 g of paracetamol before the vaccine inoculation and then every 6 h in the first 24 h. Among the 1356 individuals who received at least one dose of the vaccine at the center, 403 individuals were enrolled, with only one dose of the vaccine required as an inclusion criterion. Patients were recruited when the first dose was administered. Subsequently, they were called by telephone after the first and the second administrations in order to collect data concerning the use of paracetamol. Four parameters were considered as outcome variables, representing the most common symptoms reported after SARS-CoV-2 vaccination by scientific authorities: fever, headache, localize pain and asthenia. No evaluation scales were used, so the parameters taken into consideration are purely subjective. Information regarding this study was collected through forms filled out directly by the volunteers and a subsequent telephone recall. The data acquired included the following: sex, date of birth/vaccination, use of paracetamol and the four parameters examined.

### 2.2. Statistical Analysis

Statistical analysis was carried out using frequency and contingency tables. Concerning the univariate analysis, the differences in results between those who took and those who did not take paracetamol were evaluated through the Chi-square test for the dichotomous variables (fever, headache, localized pain, asthenia), and the nonparametric Mann–Whitney U test was applied for comparisons for the quantitative variable (number of symptoms). Given that the study is not randomized, a further multivariate analysis was carried out to consider potential confounding factors. In particular, a multiple logistic regression model was used for the dichotomous outcome variables. In this case, the results of the multivariate analysis are presented in the form of Odds ratios (OR) and 95% confidence intervals (95% CI). Moreover, a multiple linear regression model was used for the quantitative outcome variable (number of symptoms). In this case, the results are presented in the form of the β coefficient (*p*). In both types of multivariate models, the variables included in the models were the following: gender, age, type of profession. All analyses were performed using SPSS for Windows (Statistical Package for the Social Sciences, Version 27; SPSS, Inc., Chicago, IL, USA). The goodness of fit of the different linear regression models performed was evaluated using the R^2^ statistic. The significance threshold was set at *p* < 0.05 for all analysis.

## 3. Results

A team composed of two trained nurses worked at the vaccination center conducted the interviews. At the beginning of the study, they used a form (attachment A) that they personally filled out during telephone interviews with the volunteers. This form was used to investigate the four symptoms (headache, asthenia, localized pain, fever) after receiving two doses of the vaccine and, when possible, after receiving the booster shot along with the flu vaccination. Subsequently, the form was replaced with a clearer and more concise one (Attachment B) that could be filled out by the volunteers themselves during the third vaccination. New questions were introduced in the new form, such as whether paracetamol was taken and whether there was a previous COVID-19 infection. Consent to participate in the study was requested at the time of vaccination and reconfirmed in the various telephone recalls; however, some individuals did not confirm availability. Eligible individuals received at least one dose of Pfizer-BioNTech vaccine. A total of 403 volunteers were involved in the study, with 391 of them receiving two doses and twelve individuals only one dose of the vaccine.

The CONSORT diagram shows the enrollment of subjects in Figure 1.

Females represent the majority of participants (53.1%). The median age of the participants was 49.7 years.

The majority of the sample was represented by 107 (26.6%) nurses followed by 60 (14.9%) physicians. The inclusion criteria for the two study groups were based solely on the use or non-use of paracetamol and did not take into account the participants’ original professions, resulting in heterogeneity across the groups.

The characteristics of the sample are shown in Table 1.

The majority of those who took paracetamol during the first administration also took it during the second administration, while a small percentage of volunteers took paracetamol only during the second vaccination.

Specifically, 33.3% of the volunteers took paracetamol after the administration of the first dose, and this percentage increased to 40.2% after the second. The volunteers were divided as follows: some took paracetamol in the first and second vaccine administrations, some took it only in the second administration, and others who received only the first dose took it at that time.

### 3.1. First Administration

As can be seen in Table 2, 366 individuals reported no fever, and among them, 122 had taken paracetamol. On the other hand, 37 volunteers reported having a fever, and out of those, 12 had taken paracetamol. There were no statistically significant differences observed (*p* = 0.912).

In the case of headaches, 74 individuals reported experiencing this symptom, with 23 of them having taken paracetamol. On the other hand, 329 individuals did not experience headache, and among them, 111 took paracetamol, with no statistically significant differences (*p* = 0.400).

Regarding localized pain, 220 volunteers indicated having this symptom, and 75 of them took the drug. Conversely, 183 volunteers denied having this symptom, and among them, 59 took the drug, with no statistically significant differences (*p* = 044).

However, for asthenia, Table 2 shows a statistically significant association. Among the 323 individuals who reported no asthenia, 117 had taken the drug. In contrast, 80 volunteers indicated experiencing this side effect, but only 17 of them took the recommended therapy (*p* = 0.011).

It is not possible to state that paracetamol reduced the onset of total symptoms in the first administration of the vaccination because the median (range) of the number of symptoms does not show statistically significant differences.

### 3.2. Second Administration

The data illustrated in Table 3 showed similar results compared to the first administration.

In the second administration, 321 individuals reported no fever, and among them, 138 took paracetamol. On the other hand, 69 volunteers reported having a fever, with 24 of them having taken paracetamol. However, there were no statistically significant differences observed (*p* = 0.209).

Regarding headaches, 88 individuals reported experiencing this symptom, and 33 of them had taken paracetamol. Conversely, 301 individuals did not experience headaches, and among them, 128 had taken paracetamol. There were no statistically significant differences (*p* = 0.400).

Concerning localized pain, 235 volunteers wrote yes on the questionnaires, among them, 94 took the drug. In contrast, 155 wrote no, among them, 68 took the drug. There were no statistically significant differences (*p* = 0.448).

The results obtained with the first administration were consistent with those seen during the second administration, highlighting that the data of Asthenia have statistically significant association (*p* = 0.011). Among the 246 individuals who reported no asthenia, 127 took the drug. In contrast, 144 volunteers indicated experiencing this side effect, but only 35 of them took the recommended therapy.

### 3.3. Multivariate Analyses

As stated in the Methods section, two different analyses were carried out: a multiple logistic model with dichotomous symptoms as outcome variables, and a multiple linear regression analysis with the number of reported symptoms as the outcome variable. In the latter case, results are presented using beta coefficients (*p*-values).

The logistic regression analysis confirms that the use of paracetamol at both the first and second administrations time is associated with a lower probability of experiencing asthenia, with a significant reduction between 60% and 70%. There is a tendency of a reduction in fever, headache in both periods, but these results are not statistically significant. The results are shown in Table 4.

The multiple linear regression analysis confirms what emerged from the univariate analysis, i.e., the number of symptoms is inversely associated with the use of paracetamol at the second time (*p* = 0.003), as seen in Table 5.

## 4. Discussion

This study was conducted to investigate the effect of paracetamol in preventing the adverse effects of COVID-19 vaccination. The population examined consisted of 403 employees of the San Giovanni Hospital and included only healthcare workers who received the Pfizer-Biontech vaccination. The main results demonstrated that prophylactic therapy influenced the lower onset of asthenia in the first and second vaccine doses, with the reminder that each measurement is based on sensations reported by the patients. However, there were no significant differences between the two groups in terms of other side effects, including fever, headache and localized pain. It is interesting to note that the prophylaxis therapy obtained statistically significant data regarding the total number of symptoms in the second vaccination, indicating a reduction in the onset of examined symptoms. Considering the scientific literature, few studies have evaluated in an experimental or observational way the effect of paracetamol on post-vaccination COVID-19 symptoms or its effect compared to other non-steroidal anti-inflammatory drugs (NSAIDs). One such study was carried out by Kazama et al. In their research, these authors studied the effect of NSAIDs versus paracetamol to mitigate the side effect of the Pfizer/BioNTech vaccination. They evaluated 231 Japanese individuals from Miyagy University, examining symptoms such as fever, headache, and asthenia. The results showed that the duration of symptoms was shorter in volunteers who took NSAIDs compared to those who took paracetamol [8]. However, we need to consider that, unlike our study, Kazama et al. did not focus their attention on the prophylactic use of the drugs. Another study carried out in the UK by Pedro M. Folgoretti et al. considered the ChAdOx1n CoV-19 vaccine and MenACWY as a control group. As the authors stated, local and systemic reactions were more common in the ChAdOx1 nCoV-19 group, and many of these reactions were reduced with the prophylactic use of paracetamol, including pain, feeling of fever, chills, muscle aches, headache and malaise [9]. In line with Folgoretti et al., a Chinese study confirmed that the use of paracetamol in a prophylactic way reduces the risk of side effects after ChAdOx1 vaccination and declared no association with a reduction in immune response [10]. Moreover, the majority of studies in the literature concerning paracetamol after vaccination mainly focus on the hypothesis that it may induce a reduction in the antibody response to vaccines. Many of these studies involve pediatric vaccines such as the common PVC [11] and DTPa -HBV-IPV/Hib [12]. The authors concluded that although febrile reactions are significantly decreased, the administration of antipyretic drugs as prophylaxis at the time of vaccination should not be routinely recommended, as antibody responses may be reduced. Few studies, on the other hand, set the goal of establishing whether paracetamol exerts similar effects in adults. Anne M. C. M. Doedée analyzed the effect of prophylactic and therapeutic treatment with paracetamol during hepatic B vaccination, in correlation with its action on the antibody response. In this study, the authors concluded, following the analysis of the data, that only the prophylactic treatment with paracetamol and not therapeutic use had a negative influence on the antibody concentration after vaccination against hepatitis B in adults [13]. The analysis of the scientific literature suggests that few studies have been carried out on the use of paracetamol during vaccination, both in therapeutic and prophylactic ways, in adults. Therefore, its use can neither be recommended nor contraindicated. From the review of the literature, there are no studies that have shown that the use of over-the-counter and short-term doses of analgesics and antipyretics in general suppresses the immune response induced by COVID-19 vaccination. It is important to point out, however, that the assessments carried out concern almost exclusively the humoral response and not the cellular immune response [13]. Since paracetamol is generally considered safe, besides the hepatotoxic effects at higher doses, and is abundantly used as an over-the-counter drug [14], it is worth conducting studies similar to the one we are proposing.

### Strengths and Limitations

Several observational studies conducted through surveys in various European and non-European countries have shown the potential of paracetamol to reduce the side effects of different types of vaccination. The main strength of this clinical trial, even if not randomized, is that it is one of the first studies carried out on a population of workers that includes healthcare professionals with the aim of evaluating the impact of paracetamol with a prophylactic purpose after a COVID-19 Pfizer-Biotech vaccination. Previous studies conducted on this subject have focused mainly on the therapeutic effects, especially in a pediatric target.

Nevertheless, we need to acknowledge that this study has some potential limitations that need to be considered carefully. First of all, as stated before, the chosen study design is not a randomized clinical trial, which is considered as the gold standard for a clinical study. Moreover, another possible limitation of this study is related to the lack of data on the participants, apart from socio-demographic information and job titles. All the surveys took place over a year since the first volunteers were vaccinated in January 2021. This may make the data collected less reliable, as many may not fully remember the symptoms experienced, such as the different symptoms in relation to the amount of paracetamol taken. Therefore, we are not able to exclude the possibility that a recall bias, which refers to the error in remembering a particular factor by those who have to provide data on something that happened in the past, could have occurred. However, it is likely that these kinds of errors are randomly distributed in the two groups of volunteers, so this kind of bias, in general, should be relatively limited. From the review of the literature, there are no studies that have shown that the use of over-the-counter and short-term doses of analgesics and antipyretics in general suppresses the immune response induced by COVID-19 vaccination. It is important to point out, however, that the assessments carried out concern almost exclusively the humoral response and not the cellular immune response [12].

## 5. Conclusions

We can conclude that according to the data from this study, it seems that paracetamol has a positive impact on the side effects of COVID-19 vaccination, reducing asthenia in both doses and mitigating the total number of symptoms during the second vaccination. These results are extremely promising, since there is a high probability that COVID-19 vaccination will be administered each year, similar to the current flu vaccination. From a safety point of view, paracetamol is generally regarded as safe, except for the hepatotoxic effects at higher doses, and is abundantly used as an over-the-counter drug. Considering the data obtained, it would be desirable that further studies to be carried out on the topic under consideration, including the implementation of a randomized clinical design.

## Figures and Tables

**Figure 1 vaccines-11-01493-f001:**
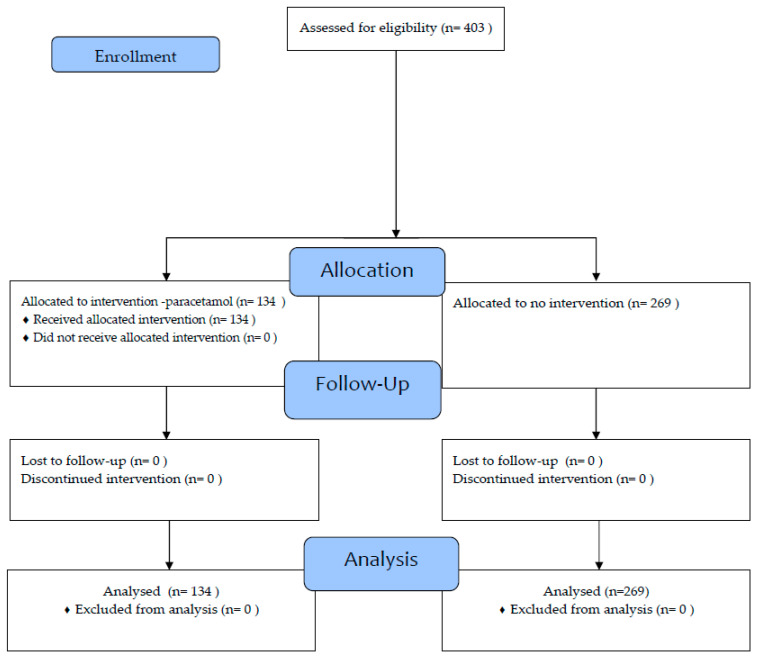
Flow-diagram of the participants in the clinical trial.

**Table 1 vaccines-11-01493-t001:** Characteristics of the sample.

Variables	n (%) or Median (Range)
*Gender*	
Female	214 (53.1%)
Male	189 (46.9%)
*Age (years)*	49.7 (18.5–103)
*Role*	
Nurses	107 (26.6%)
Physicians	60 (14.9%)
Other healthcare professionals	22 (5.5%)
Administratives	53 (13.2%)
Other	76 (19.9%)
Health Care Assistant	41 (13.2%)
Physiotherapists	44 (10.9%)

Column 2 shows the incidence (n) and proportions (%) of respondents with a given characteristic (row) among the total sample.

**Table 2 vaccines-11-01493-t002:** Difference of symptoms between those who took paracetamol and those who did not—first period.

Symptoms	Paracetamol Yes n (%) n Tot = 134	Paracetamol NO n (%) n Tot = 269	*p*
*Fever*			
No	122 (91%)	244 (90.7%)	0.912
Yes	12 (9.0%)	25 (9.3%)	
*Headache*			
No	111 (82.8%)	218 (81%)	0.661
Yes	23 (17.2%)	51 (19%)	
*Localized pain*			
No	59 (44.0%)	124 (46.1%)	0.695
Yes	75 (56.0%)	145 (53.9%)	
*Asthenia*			
No	117 (87.3%)	10 (76.6%)	0.011
Yes	17 (12.7%)	63 (23.4%)	
*Number of Symptoms*	1 (0–4)	1 (0.4)	0.314

Columns 2 and 3 show the incidence (n) and proportions (%) of respondents with the symptoms given in that row, showing the designated results in each column. Comparisons involve the proportions of respondents who did or did not take the drug, showing a designated outcome. Evaluations were performed using chi-square analysis.

**Table 3 vaccines-11-01493-t003:** Difference of symptoms between those who took paracetamol or those who did not—Second period.

Symptoms	Paracetamol Yes n (%) n Tot = 134	Paracetamol NO n (%) n Tot = 269	*p*
*Fever*			
No	138 (85.2%)	183 (80.3%)	**0.209**
Yes	24 (14.8%)	45 (19.7%)	
*Headache*			
No	128 (79.5%)	0.209	0.400
Yes	33 (20.5%)	55 (24.1%)	
*Localized pain*			
No	68 (42.0%)	87 (38.2%)	0.448
Yes	94 (58.0%)	141(61.8%)	
*Asthenia*			
No	127 (78.4%)	119 (52.2%)	**0.001**
Yes	35 (21.6%)	109 (47.8%)	
*Number of Symptoms*	1 (0–4)	2 (0–4)	**0.001**

Columns 2 and 3 show the incidence (n) and proportions (%) of respondents with the symptoms given in that row, showing the designated results in each column. Comparisons involve the proportions of respondents who did or did not take the drug, showing a designated outcome. Evaluations were performed using chi-square analysis.

**Table 4 vaccines-11-01493-t004:** Results of the Multiple logistic regression analysis.

Outcome Variables	Paracetamol 1 OR (95% CI)	Paracetamol 2 OR (95% CI)
*Fever*	0.845 (0.55–1.31)	0.72 (0.42–1.24)
*Headache*	0.92 (0.64–1.33)	0.86 (0.52–1.41)
*Localized pain*	1.09 (0.80–1.47)	0.92 (0.60–1.40)
*Asthenia*	0.42 (0.29–0.60)	0.30 (0.18–0.48)

**Table 5 vaccines-11-01493-t005:** Multiple linear regression analysis, the dependent variable: number of symptoms.

	Time 1 β (*p*)	Time 2 β (*p*)
*Paracetamol*	−0.026 (0.596)	−0.146 (0.003)
*R2 of the model*	0.059	0.105

## Data Availability

Data are available upon request.

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
