# Peer review of "The Effectiveness of Paracetamol to Reduce the Post-Vaccination SARS-CoV-2 Adverse Effects in an Italian Vaccination Center"

_vaccines, 2023, doi:10.3390/vaccines11091493_

Round 1
Reviewer 1 Report
The study written by Ricci E et al., entitled "The effectiveness of paracetamol to reduce the post-vaccination SARS-CoV-2 adverse effects in an Italian Vaccination center" aims to analyze the impact of the prophylactic use of paracetamol to reduce the post-vaccination Comirnaty/Pfizer adverse effects. This study took place at the San Giovanni Battista Hospital in Rome and included all hospital employees who received two doses of Pfizer-BioNTech. The vaccination health personnel recommended the preventive intake of 1g paracetamol before inoculation of the vaccine and then every 6 hours in the first 24 hours. Information was collected through forms and a subsequent telephone recall. A total of 403 volunteers were involved in the study, 391 of them received two doses and 12 people only one dose of the vaccine. The main results demonstrated that the prophylactic therapy influenced the lower onset of asthenia in the first and second dose. However, there are no significant differences between the two groups in terms of fever, headache and localized pain. Paracetamol has a good impact on the side effect of Covid-19 Vaccination, reducing asthenia in both doses and mitigating the total of symptoms during the second vaccination. The results are presented clearly. However, before the work is published, I suggest enriching the article with information on the concentration of antibodies before and after vaccination in order to assess whether the administration of paracetamol did not reduce the effectiveness of vaccination.
Author Response
The study written by Ricci E et al., entitled "The effectiveness of paracetamol to reduce the post-vaccination SARS-CoV-2 adverse effects in an Italian Vaccination center" aims to analyze the impact of the prophylactic use of paracetamol to reduce the post-vaccination Comirnaty/Pfizer adverse effects. This study took place at the San Giovanni Battista Hospital in Rome and included all hospital employees who received two doses of Pfizer-BioNTech. The vaccination health personnel recommended the preventive intake of 1g paracetamol before inoculation of the vaccine and then every 6 hours in the first 24 hours. Information was collected through forms and a subsequent telephone recall. A total of 403 volunteers were involved in the study, 391 of them received two doses and 12 people only one dose of the vaccine. The main results demonstrated that the prophylactic therapy influenced the lower onset of asthenia in the first and second dose. However, there are no significant differences between the two groups in terms of fever, headache and localized pain. Paracetamol has a good impact on the side effect of Covid-19 Vaccination, reducing asthenia in both doses and mitigating the total of symptoms during the second vaccination. The results are presented clearly. However, before the work is published, I suggest enriching the article with information on the concentration of antibodies before and after vaccination in order to assess whether the administration of paracetamol did not reduce the effectiveness of vaccination.
Answer: many thanks for this suggestion. From the review of the literature there are no studies that have shown that the use of over-the-counter and short-term doses of analgesics and antipyretics in general suppresses the immune response induced by Covid vaccination. It is important to point out, however, that the assessments carried out concern almost exclusively the humoral response and not the cellular immune response. (reference 12)
Reviewer 2 Report
Dear authors,
Congratulations for your study!
After reading the article my comments / suggestions are the following:
- lines 11-12 and lines 50-51- please give the reason for which you consider that the side effects would limit the adherence to COVID-19 vaccines especially in health care workers;
- line 34 - explain what DPI means;
- line 69 - I would remove “In conclusion“ from the beginning of the paragraph;
- Figure 1 - should be mentioned clearly that those from the first arm received the intervention (paracetamol) and those from the second arm did not;
- Table 1 - explain what OSS means.
The English language is fine.
Author Response
Dear authors,
Congratulations for your study!
Answer: thansks for your comment.
After reading the article my comments / suggestions are the following:
- lines 11-12 and lines 50-51- please give the reason for which you consider that the side effects would limit the adherence to COVID-19 vaccines especially in health care workers;
Answer: many thanks for this comment. In fact, as demonstrated by the cross-sectional survey of HCWs carried out in California, among the 2103 vaccinated, 34% experienced side effects during work, 28% had to interrupt their work and 11% declared that the symptoms interfered moderately a strict requirement for the correct fulfillment of professional responsibilities. So its can lead to absenteeism, and impact future vaccine decision-making.
- line 34 - explain what DPI means;
Answer: done. Now we specify this is personal protection equipment (PPE)
- line 69 - I would remove “In conclusion“ from the beginning of the paragraph;
Answer: done as requested
- Figure 1 - should be mentioned clearly that those from the first arm received the intervention (paracetamol) and those from the second arm did not;
Answer: done as requested
- Table 1 - explain what OSS means.
Answer: done as requested. Now we specify this is “Health Care Assistant”
Reviewer 3 Report
Authors have done a study to compare the role of paracetamol in preventive two outcome variables among those who took a second dose of the Pfizer COVID-19 vaccine. Although the study is important but not written properly, there are a number of loosely arranged and incomplete sections in the manuscript that need to be strengthened.
Why do authors specify that only healthcare workers will be affected due to the adverse effects of COVID-19 vaccines? This has been included in line 11. Maybe you can specify why it’s important in healthcare workers, the adverse effect profile of the vaccine is important.
The Figure 1 CONSORT diagram is not complete.
Include more description of each cohort included in the individual arm, their type like nurses, physicians, and so on.
In the follow-up section, there is no need to write ‘give reasons’ as there is no exclusion.
In Table 1, why it is written as ‘n (%) or median (range)’, I think the values are given in n (%) that only have to be specified.
Table legend should be included in each table to explain how the data was expressed, comparisons made with the name of the test/s that is applied, and expansion of abbreviations used.
How asthenia was measured in the study group?
Did the author experience any difference between the group that took two paracetamol tablets compared to those who took one?
Authors should be clear in their inclusion criteria. Was included with or two doses of vaccine, need to consistent throughout, in section 2.1, it is contradicting.
How samples were recruited, ethical issues, approval number and informed consent need to be discussed in section 2, methods, missing completely.
It is very confusing to understand whether patients were recruited after two doses or when they above to take the second dose or third dose. In the result, the scenario is different. It says about enclosures A and B. It means you have both classes, if that is the case, that should be included in the flow chart and compare the outcome of those two groups.
Overall, I suggest authors conduct a proof of the whole manuscript to eliminate many contradictory statements.
moderate correction is needed.
Author Response
Authors have done a study to compare the role of paracetamol in preventive two outcome variables among those who took a second dose of the Pfizer COVID-19 vaccine. Although the study is important but not written properly, there are a number of loosely arranged and incomplete sections in the manuscript that need to be strengthened.
Why do authors specify that only healthcare workers will be affected due to the adverse effects of COVID-19 vaccines? This has been included in line 11. Maybe you can specify why it’s important in healthcare workers, the adverse effect profile of the vaccine is important.
Answer: Many thanks for this comment. We added the following issue. In fact, as demonstrated by the cross-sectional survey of HCWs carried out in California, among the 2103 vaccinated, 34% experienced side effects during work, 28% had to interrupt their work and 11% declared that the symptoms interfered moderately a strict requirement for the correct fulfillment of professional responsibilities. So its can lead to absenteeism, and impact future vaccine decision-making.
The Figure 1 CONSORT diagram is not complete.
Include more description of each cohort included in the individual arm, their type like nurses, physicians, and so on.
Answer: Many thanks for this suggestion. However, the inclusion criteria for the courts in the two arms did not take into account the profession of origin but exclusively the intake or not of paracetamol, for this reason the courts are heterogeneous.
In the follow-up section, there is no need to write ‘give reasons’ as there is no exclusion.
Answer: we agree woth the reviewer, and removed this from all the figure
In Table 1, why it is written as ‘n (%) or median (range)’, I think the values are given in n (%) that only have to be specified.
Answer: we leave the median (range) due to the variable age.
Table legend should be included in each table to explain how the data was expressed, comparisons made with the name of the test/s that is applied, and expansion of abbreviations used.
Answer: Many thanks for this suggestion
Table 1: Column 2 shows the incidence (n) and proportions (%) of respondents with a given characteristic (row) among the total sample
Table:2
- Columns 3 and 4 show the incidence (n) and proportions (%) of respondents with the symptoms given in that row showing the designated result of each column.
- Comparisons involve the proportions of respondents who did or did not take the drug who show a designated outcome. Evaluations were performed using chi-square analysis. Statistically significant differences: * p < 0.05, ** p < 0.01, *** p < 0.001.
.
How asthenia was measured in the study group?
Answer: each measurement is based on sensations reported by the patients.
Did the author experience any difference between the group that took two paracetamol tablets compared to those who took one?
Answer: No it’s considered as a bias such as the different symptoms in relation to the amount of paracetamol taken
Authors should be clear in their inclusion criteria. Was included with or two doses of vaccine, need to consistent throughout, in section 2.1, it is contradicting.
Answer: only one dose of vaccine was required as an inclusion criterion
How samples were recruited, ethical issues, approval number and informed consent need to be discussed in section 2, methods, missing completely.
Answer: the recruitment process is described in the Methods, paragraph “Study Design and setting”.
As explained at page 9, since the study was an observational study, the approval of the Ethical Committee is not requested. However, the study was carried out following the Helsinki declaration. Informed consent was obtained from all subjects involved in the study, and the form was submitted to the Editorial staff
It is very confusing to understand whether patients were recruited after two doses or when they above to take the second dose or third dose. In the result, the scenario is different. It says about enclosures A and B. It means you have both classes, if that is the case, that should be included in the flow chart and compare the outcome of those two groups.
Answer: we agree with the reviewer about possible confusion. We now specify: “patients were recreuited when the first dose was administered. Subsequently they were called by telephone after the first and the second administration, in order to collect data concerning the use of paracetamol.”
Overall, I suggest authors conduct a proof of the whole manuscript to eliminate many contradictory statements.
Answer: done as requested.